# Complex hemolymph circulation patterns in grasshopper wings

Mary K. Salcedo [1✉], Brian H. Jun [2], John J. Socha [3], Naomi E. Pierce [4], Pavlos P. Vlachos [5] &
Stacey A. Combes[6]

An insect's living systems—circulation, respiration, and a branching nervous system—extend from the body into the wing. Wing hemolymph circulation is critical for hydrating tissues and supplying nutrients to living systems such as sensory organs across the wing. Despite the critical role of hemolymph circulation in maintaining healthy wing function, wings are often considered "lifeless" cuticle, and flows remain largely unquantified. High-speed fluorescent microscopy and particle tracking of hemolymph in the wings and body of the grasshopper *Schistocerca americana* revealed dynamic flow in every vein of the fore- and hindwings. The global system forms a circuit, but local flow behavior is complex, exhibiting three distinct types: pulsatile, aperiodic, and "leaky" flow. Thoracic wing hearts pull hemolymph from the wing at slower frequencies than the dorsal vessel; however, the velocity of returning hemolymph (in the hindwing) is faster than in that of the dorsal vessel. To characterize the wing's internal flow mechanics, we mapped dimensionless flow parameters across the wings, revealing viscous flow regimes. Wings sustain ecologically important insect behaviors such as pollination and migration. Analysis of the wing circulatory system provides a template for future studies investigating the critical hemodynamics necessary to sustaining wing health and insect flight.

[1] Department of Biological and Environmental Engineering, Cornell University, Ithaca, NY, USA. [2] School of Mechanical Engineering, Purdue University, West Lafayette, IN, USA. [3] Department of Biomedical Engineering and Mechanics, Virginia Tech, Blacksburg, VA, USA. [4] Department of Organismic and Evolutionary Biology and Museum of Comparative Zoology, Harvard University, Cambridge, MA, USA. [5] Weldon School of Biomedical Engineering, Purdue University, West Lafayette, IN, USA. [6] Department of Neurobiology, Physiology and Behavior, UC Davis, Davis, CA, USA. ✉email: mks276@cornell.edu

nsect wings are often thought of as dead, lifeless cuticle, but a functioning and healthy wing is inextricably linked to active circulatory flow within[1–3]. Hemolymph, an insect's blood, serves to hydrate tissues, supply nutrients to the nervous and respiratory systems, and circulate cells involved in immune function, providing critical physiological function in all insects[4–6]. These systems extend and branch into the wing as well, necessitating nourishment by hemolymph, as in the body[7,8]. Flow of hemolymph is also involved in insect development, serving as a hydraulic tool during growth, metamorphosis, eclosion, and wing expansion[9,10]. Within the wing itself, hemolymph circulation is necessary for living organs and sensory structures[11], such as scent-producing organs on lepidopteran wings[8] and thousands of sensory hairs distributed across dragonfly wings[6,12,13]. Structural tissues embedded in wing veins, like resilin[14,15], depend on hemolymph hydration, providing the wetted, living wing with different mechanical properties than a dried, dead wing, demonstrating the essential role of circulation for insect flight[16]. In fact, under desiccation, insect cuticle dramatically decreases in toughness[17]. However, while the structural and aerodynamic properties of insect wings are relatively well-studied[18], the internal, living systems within wings—and the flow that supplies them—have been largely ignored, despite their critical importance for insect ecology and evolution.

Some broad trends concerning circulation in wings are well understood. Across 14 insect orders, there are two main flow patterns in resting insects: circuitous flow (one-way: circuit-like) and tidal flow (two-way: in all veins at once, and then out)[7,19,20]. Mosquito wings, for example, exhibit circuitous flow within their tiny millimeter-scale wings, driven by an independent thoracic wing heart that pulls hemolymph from the wing in a pulsatile fashion[21]. Lepidopterans, in contrast, demonstrate tidal flow in some species; the giant Atlas moth (*Attacus atlas*), with a wing span of 30 cm, uses multiple thoracic wing hearts, thoracic air sacs, and tracheae extending into the veins to push and then pull hemolymph through all wing veins[19]. Recent work in smaller lepidopterans revealed tidal flow in one species (*Vanessa cardui*) but circuitous flow in two others (*Satyrium caryaevorus* and *Parrhasius m-album*) with scent-producing organs in their wings[8], suggesting that flow patterns may function to service specific wing structures.

However, quantitative analyses of hemolymph circulation within wings are still scarce, particularly those identifying local flow behaviors within the veins. Previous studies have focused on qualitative or bulk-flow measurements in insects at rest[12,20,22]. Measuring fluid movement within insect wing veins is a difficult task, even in a stationary wing[1]. In the last decade, increased use of injected fluorescent dye or particles and high-speed video has allowed for more detailed hemolymph flow measurements in resting insects[1,12,21,23,24]. Such tools have revealed that in the mosquito *Anopheles gambiae*, hemolymph enters the wing at a slower speed (99 μm per second), returning to the body at a much faster speed (458 μm per second), a difference of ~4.5×[21]. Increased computational resources have enabled dynamic 3D models of flow in flapping wings, which suggest that the presence of hemolymph reduces aerodynamic instabilities like flutter[25]. Recently, combined experimental and analytical modeling has shown that flapping can induce faster hemolymph flows within the wing than those observed during rest[12], but the method led to high mortality and low activity in the majority of the insects.

Furthermore, the circulatory network itself is not a simple system of uniform pipes. Vein size can vary dramatically within a wing, tapering both from base to tip (across the span) and from edge to edge (across the chord), as well as across species, with diameter differing by almost three orders of magnitude from 0.5 μm in mosquitoes to 300 μm in large moths. Additionally, some regions of the wing are supplied with hemolymph via routes that do not have clear channel-like structures provided by wing veins (i.e. leaky membrane regions)[7,26]. The circular scent-producing patches in the wings of lycaenid butterflies (*Eumaeini*)[8], for instance, may provide a porous resistance to flow, and the dragonfly's pterostigma, a discrete, rectangular sinus at the wing tip, is thought to hold copious amounts of hemolymph[26]. Such elements increase the complexity of circulation in the wing and preclude simple mechanical modeling of the system. Models not only provide fundamental insight into physiological function, but they are also necessary for bio-inspired or biomimetic engineering to produce more effective microfluidics, important in a broad range of applications including biosensors, drug-delivery devices, and lab-on-a-chip devices[27–29].

Overall, there is a lack of understanding of specific flow patterns inside wing veins and how they relate to the geometry of the wing's circulatory network. This gap in knowledge, between description and quantitative flow, hinders our understanding of the important role hemolymph plays in insect behavior and healthy wing function.

Here, we used high-speed, fluorescent particle microscopy to observe, track, and quantify active hemolymph circulation within the densely venated fore- and hindwings, as well as the body of live, adult American bird grasshoppers *Schistocerca americana* at rest. We chose this species because of its history as a model organism for studies of flight; *S. americana* is known for its damage to agriculture, and its wings, and those closely related to it, have been investigated in terms of wing biomechanics, damage, and structural characteristics[30–33]. Adult specimens are of intermediate size and have more complex and dense wing venation relative to previous wing circulation studies (e.g. the mosquito and Atlas moth). As an ecologically relevant pest species, *S. americana* are readily available through collaborations with U.S. Department of Agriculture (USDA) facilities.

We investigated two hypotheses based on previous knowledge of mosquitoes, the only other insect for which flow rates are known[21]: (1) hemolymph traverses every vein within a venation network, (2) hemolymph returns to the body through trailing edge veins at a velocity much faster than that entering the wing. Given the numerous unknowns concerning how structures embedded in the wing or venation patterns interact with hemolymph flow, we also explored two additional questions: (3) do flow patterns change depending on location within the wing? and (4) what is the involvement of flow outside of enclosed veins, within amorphous structural sinuses? Lastly, we characterized the dynamics of varying fluid behavior by calculating the dimensionless Reynolds, Womersley, and Péclet numbers throughout the wing, providing fundamental insight into the physical rules that guide its circulatory flow.

## Results and discussion

**Hemolymph flow in every vein.** Structurally, insect wings are composed of chitinous, tubular veins and thin, membranous regions[5]. While veins are supportive structures, they are also extensions of the open circulatory and tracheal systems (Fig. 1a), driven by thoracic wing hearts which pull hemolymph through a wing (Fig. 1b). Hemolymph hydrates tissues, and veins containing tracheal tubes (Fig. 1c). The tracheal system, a branching network, serves to deliver oxygen directly to tissues throughout the body and appendages via diffusion and advection (bulk flow)[34]. Tracheal branches first extend into wing tissue during wing pad development and can be found in most, but not all adult wing veins[7,35]. Tracheae can also be seen compressing under pulses of hemolymph (Supplementary Movie 3), demonstrating a

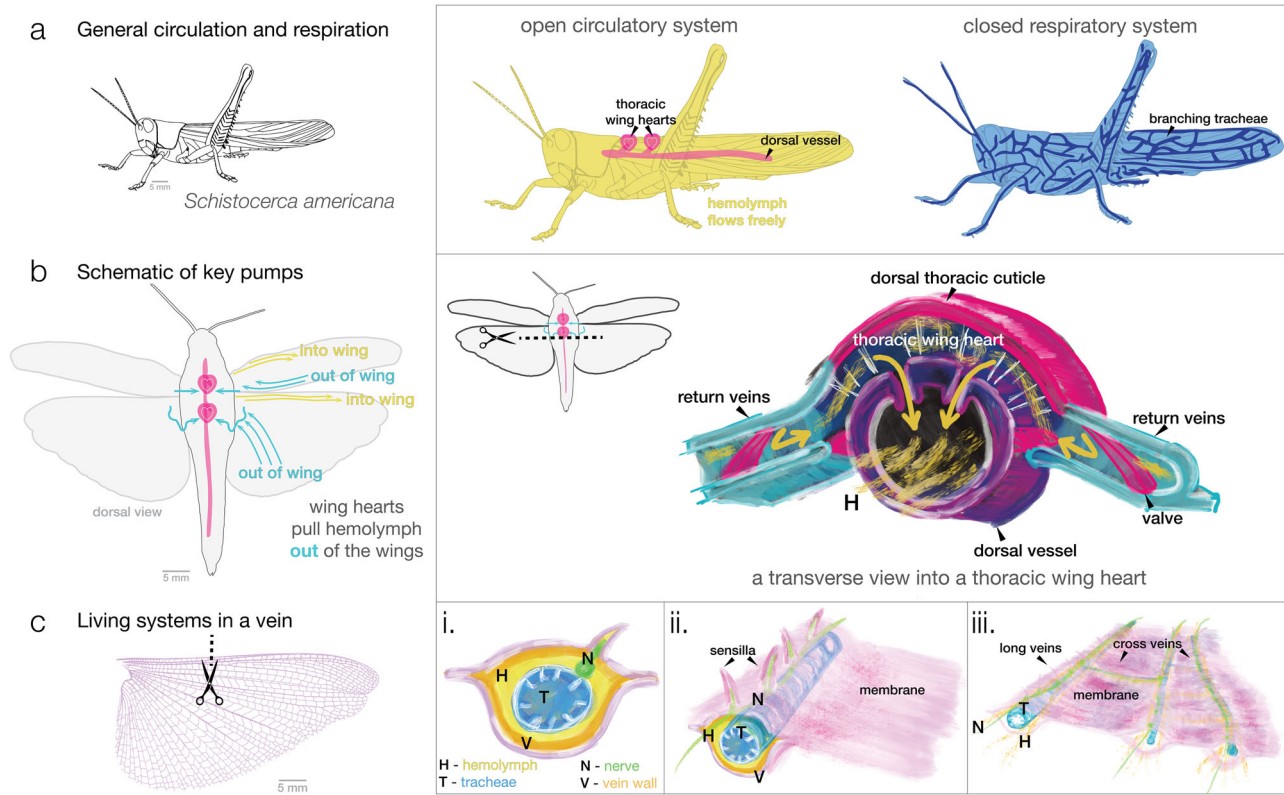

**Fig. 1 Physiology of fluid systems in the grasshopper *Schistocerca americana* and its wings. a** This cartoon features the two main fluid systems within an insect: the open circulatory system (left) and closed respiratory system (right). Within an open circulatory system, hemolymph (insect blood) is commonly pumped from posterior to anterior via a long tubular heart called the dorsal vessel. Accessory hearts (i.e., pumps) in the thorax called "wing hearts" pump blood from the wing[40]. An insect's respiratory system is a network of tracheal tubes and air sacs, which transport oxygen and carbon dioxide directly to the tissues via advection and diffusion (diagram is representative)[34]. **b** In *S. Americana*, thoracic wing hearts have "return conduits" (i.e., scutellar branches) where hemolymph leaves the wing and returns to the main heart. A transverse slice through the thorax reveals how these thoracic wing hearts are located dorsally above the main tubular heart. **c** An example cross-section (not proportional) through a vein reveals (i) hemolymph, tracheal branches, nerves, and vein wall[2]. Extended views (ii) and (iii) show nerve branches connecting to proprioceptors, and how hemolymph and tracheal tubes form networks inside the wing. Drawings in **b**, **c** inspired by Pass[2,3].

mechanical coupling between the circulatory and respiratory systems[19]. In some parts of the wing, tracheae can obstruct flow (as in Supplementary Movie 3), and when hemolymph flow reverses, tracheae can be seen expanding.

Using neutrally-buoyant fluorescent particles and high-speed video, we recorded particles flowing in sync with hemocytes being advected into the wing, through trailing edge wing veins, near the wing base (Fig. 2a–c, Supplementary Movie 1). To quantify flows, we employed tracking methods including (1) automated multiparametric particle tracking (Fig. 3a) and (2) semi-automated particle tracking in Matlab (DLTdv5)[36] to follow hundreds of particles throughout the wing and body (~800, see Supplementary Fig. 1). The time-varying position data of these tracked particles were used to calculate instantaneous flow velocities and identify flow patterns in resting *S. americana* adults (Fig. 2c). Shown in Fig. 3a are example flow paths and corresponding mean instantaneous velocity of all visible particles on a series of images measured over time (for one representative individual). Particle movements generally appear more coordinated in regions where pulsatility is dominant (i.e., wing base, Fig. 3a, bulk flow), compared to where it is not pulsatile (i.e. wing tip, Fig. 3a, aperiodic flow). In the wing schematics of Fig. 2c, particle location indicates the normalized starting position of tracking, and clustering indicates that multiple particles were tracked within a region. In Fig. 2c (bottom), the particles within the body are jittered and represent a general tracking location.

The forewing of grasshoppers, the tegmen, is a thickened, semi-leathery structure that covers the larger hindwing (2.5× greater area), which is folded like a corrugated fan under the forewing when the insect is at rest[37]. Both wings are densely venated, and contain longitudinal veins spanning from base to tip, which is interconnected by numerous, shorter cross-veins (Fig. 3b). Particularly near the wing base in the leading edge region of the forewing, the wing veins are not uniformly circular in cross-section, but appear flattened and shallow, interconnected by many S-shaped cross-veins. Particles in this region were recorded flowing alongside hemocytes (Supplementary Movie 1).

We confirmed that hemolymph traverses all veins, including cross veins and certain areas of the wing membrane, within the grasshopper wing network (Supplementary Movies 1–6), even to the edges of the wings, where veins are most likely to be damaged[38] (Fig. 3c, d).

**Local flow behaviors depend on wing location**. Vein structure and pattern across the wing influence how hemolymph moves through the wing. Because wing structure–flow interactions have not been previously investigated, we identified five distinct wing regions based on structural similarities between the wings, within which we assessed local flow characteristics (Fig. 3b). These regions include the following: (1) leading edge (largest diameter veins), (2) membrane (a large sinus present between wing layers

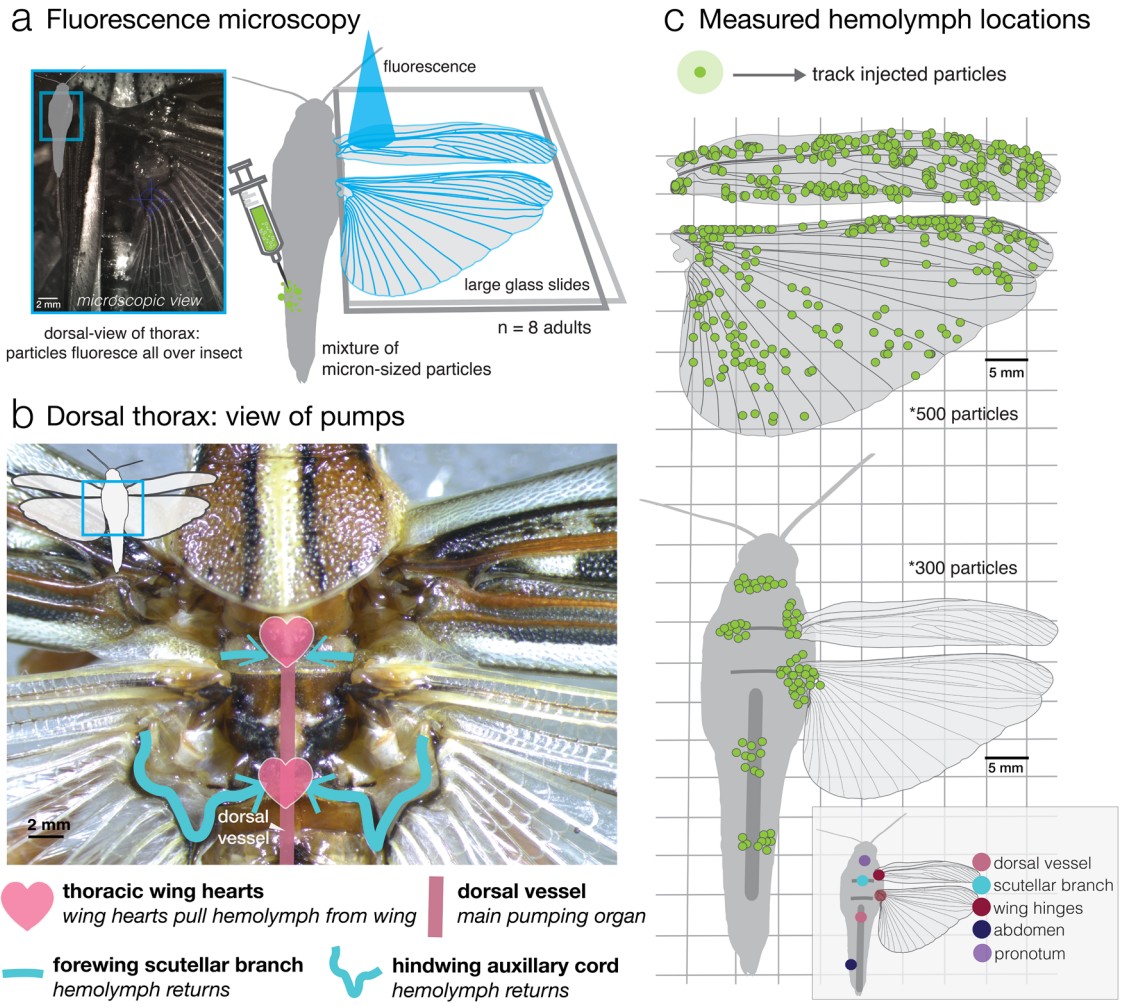

**Fig. 2 Visualizing and quantifying hemolymph flow via fluorescent particle tracking in *S. americana*. a** View of the dorsal thorax of a grasshopper under a fluorescent microscope (left). Live insects were injected with neutrally buoyant fluorescent particles. Before imaging and particle injection, *S. americana* was briefly anesthetized with carbon dioxide and quickly restrained with modeling clay; wings were spread between two glass slides (blue light indicates fluorescence). **b** Dorsal view indicates the location of thoracic wing hearts and return conduits into the wing heart pump. The dorsal vessel dominates the pumping of hemolymph within an insect, but cannot circulate hemolymph into the wings without the assistance of thoracic wing hearts, which pump hemolymph from the wing. **c** Wing map (normalized coordinates) of all particles (500 total) tracked and quantified across 8 adult grasshoppers in both the forewing and hindwing (16 wings total). Body map of measured particles (300 total) across the body.

near the leading edge), (3) wing tip (small diameter, highly interconnected veins), (4) lattice (mostly orthogonally connected veins), and (5) trailing edge (larger diameter veins) (Fig. 3b–d).

To analyze hemolymph velocity at different locations, we calculated instantaneous maximum and median particle velocities across the five wing regions (Fig. 4a, b, see methods for calculations and Supplementary Fig. 1). Overall, flow velocities are higher in the hindwing (Fig. 4a). Within each wing, the highest peak flow velocities occur in regions near the wing base; in the forewing and hindwing the highest peak velocities occur in the trailing edge region (where the thoracic wing hearts pull hemolymph out of the wing through the scutellar branch and auxiliary cord). Median flow velocities were also calculated to characterize typical flow speeds; these values show similar trends (Fig. 4b), but with smaller differences between regions (Supplementary Fig. 1).

Structurally, veins in the leading and trailing edge regions of the forewing and hindwing are wider in diameter than those in the tip, lattice, and membrane regions. At the wing tip, flow follows the perimeter vein and begins moving down the chord of the wing into the lattice and trailing edge regions. In both the

fore- and hindwings, hemolymph flow slows substantially in the wing tip region (Fig. 4a, b) and increases again in the trailing edge region. In the hindwing, the fan-like anal veins within the trailing edge region serve as long conduits (with fewer junctions to traverse) that all feed into the same return conduit (i.e., auxiliary cord), where flow is pulled out by the posterior thoracic wing heart (Fig. 2b, Supplementary Movie 6).

Although bulk flow within grasshopper wings can be described as a one-way circuit, with hemolymph entering via leading edge veins and exiting from trailing edge veins (Fig. 3c, Supplementary Movies 1 and 6), local flow behaviors within the veins are complex and time-varying. Hemolymph does not travel along simple, predetermined paths through the wing, but rather may display one of several local flow behaviors at any given vein junction, at any particular point in time (Fig. 3d). Specifically, while measuring and tracking active hemolymph circulation in every vein within both the fore- and hindwing, we observed three distinct, local flow behaviors: pulsatile, aperiodic, or leaky flow (Supplementary Movies 1–6), described in detail below. Combinations of the types of flow behaviors can be found within many of the wing regions, and the occurrence of some local behaviors

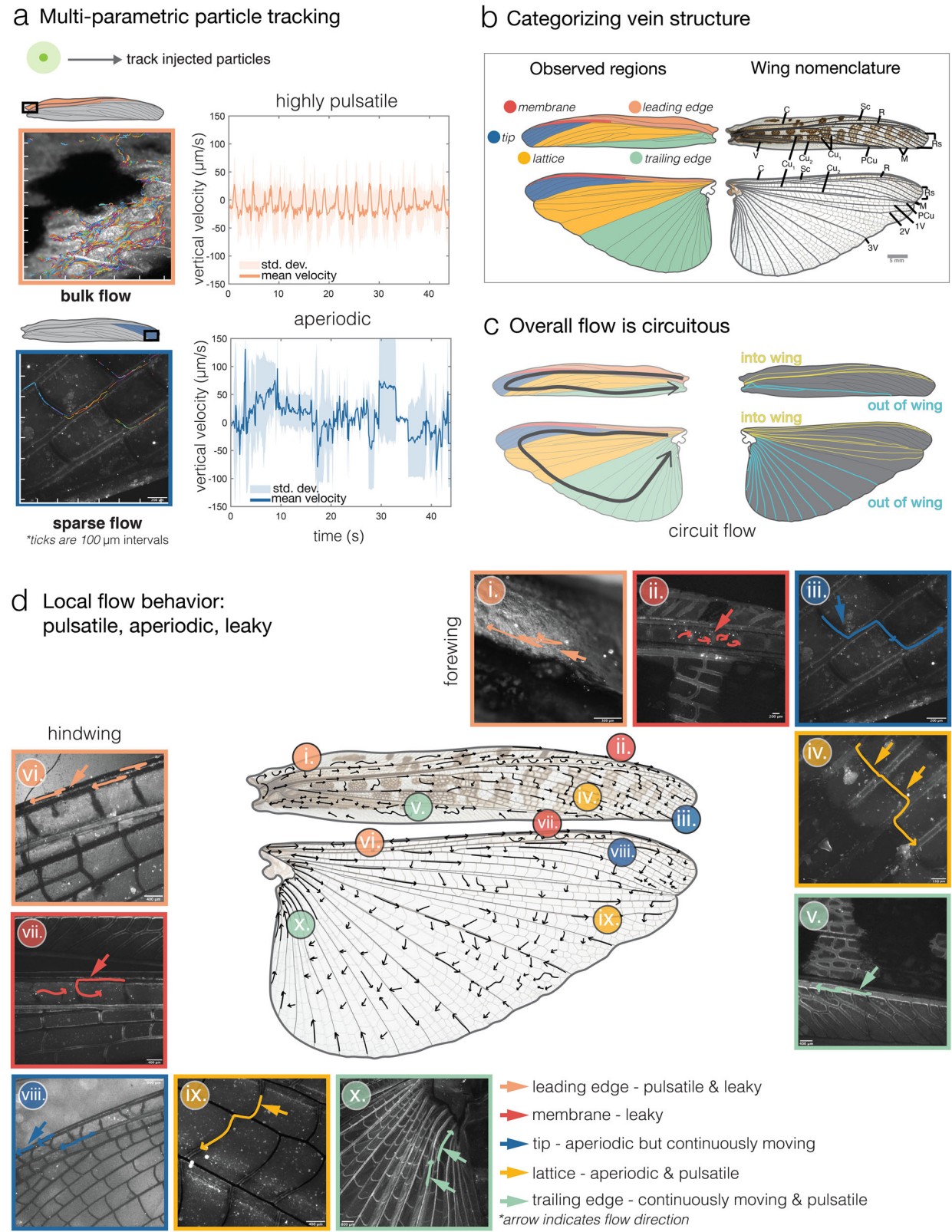

appears to be a function of proximity to the body and its associated pumping organs (Fig. 3d).

Hemolymph is pumped out of the forewing and hindwing near the trailing edge by each wing's respective thoracic wing heart (Fig. 2b, Supplementary Movie 8), and hemolymph flows into the wing from the thoracic space, entering through the largest leading edge veins, the costa, subcosta, and radius. These thoracic flows are also influenced by the respiration of thoracic air sacs. Overall, there is a circuitous pattern of flow in both the fore- and hind wings (Fig. 3c). Within the wing, hemolymph flows faster in

**Fig. 3 Patterns of circulatory hemolymph flow in the wings. a** We used multi-parametric particle tracking to detect bulk movement of particles, which allows for tracking of large numbers of particles. Flow near the wing base (bulk flow) shows distinct pulsatility (across individuals), whereas in regions such as the wing tip (lattice region), flow traverses more junctions, and patterns are less periodic (aperiodic jumps indicate travel between cross veins to long veins). These example plots show the vertical velocity (*y*-component) for hundreds of particles within a region for a representative individual (both top and bottom). **b** Wing metrics were categorized into five regions (left) based on vein location and structure: (1) leading edge (pink, costa to subcosta), (2) membrane (red, subcosta to radius), (3) wing tip (dark blue, radial sector to medius), (4), lattice (yellow, medius to post cubitus), and (5) trailing edge (light green, post cubitus to vannal region). Labels follow long-vein nomenclature (short veins are typically unnamed). **c** Overall flow in the wing was found to be circuitous, where hemolymph moved into the wing through C, Sc, and R veins, and out of the wing via the Cu and V veins[37]. **d** Following Arnold's (1964)[7] wing drawings, hand-drawn vectors represent hemolymph behavior (based on tracking analysis). Tracking of fluorescent particles reveals that flow behaves in three modes: pulsatile (double-headed arrow), leaky (curved arrow), and aperiodic (straight arrow). Examples of forewing venation (i.–v.) and hindwing (vi.–x.) in each of the five regions. Wing veins: C—costa, Sc—subcosta, R—radius, Rs—radius sector, M—medius, Cu—cubitus, PCu—post cubitus, V—vannal.

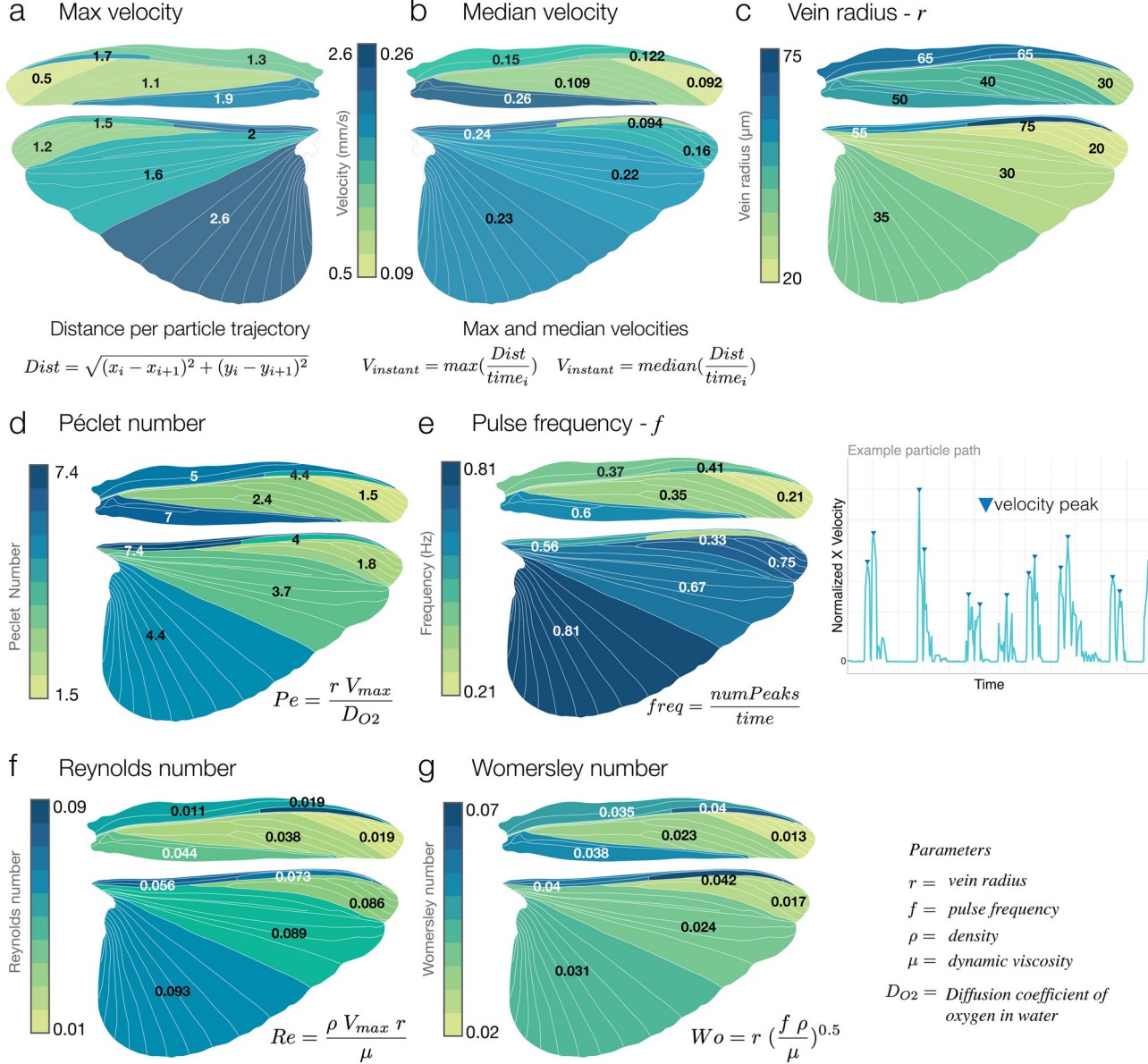

**Fig. 4 Flow dynamics across wing regions. a** and **b** Maximum and median velocities per wing region. Faster flows occur in the trailing edge of the forewing and the leading edge of the hindwing. **c** Average vein radius per region (*n* = 25 vein radii measured, and average taken). **d** Average Péclet number for each region, where the diffusion coefficient, $D_{O2}$, is for oxygen in the water. **e** Average pulse frequency, calculated by a number of velocity peaks over time (right of **e**), increases in the trailing edge. **f** Average Reynolds number, inertial to viscous forces, describes a viscous flow regime in all regions (1<). **g** Average Womersley number, pulsatility to viscous flows, describes flow similar to venules[42]. Particle trajectories were placed on a normalized wing coordinate system (*n* = 8 individual grasshoppers and 500 digitized particles).

regions near the body and slower in regions toward the tip of the wing. These higher speeds may reflect proximity to the main pumping organs, or perhaps reflect functional requirements for greater hydration at the wing hinge, correlating with the thick layers of resilin[39].

This circulatory pattern of flow is similar to that observed in mosquitoes[21]. However, the relative magnitude between the inflow (into the wing) and outflow (returning to the body) is far less skewed. Maximum flow velocities in these grasshopper wings range from 0.5 to 2.6 mm/s (Fig. 4a) with into-wing and out-of-wing ratios being 1.5 and 1.3 in the forewing and hindwing. Comparatively, the in/out ratio of flow is 4.6 in the mosquito (*A. gambiae*) at rest, where the difference may relate to thoracic wing heart structure: in mosquitoes, the thoracic wing heart is detached from the dorsal vessel[21], and operates at an independent frequency of 3 Hz, whereas, in grasshoppers, the thoracic wing hearts exist as modified dorsal vessel tissue, attached, right above the dorsal vessel (Fig. 1b)[40]. Overall, hemolymph return is much faster to the body in mosquitoes than in grasshoppers.

**Specific flow behaviors: pulsatile, aperiodic, and leaky**. Flow is pulsatile in much of the wing (Supplementary Movies 2 and 6), with particles pulsing forward and then stopping or reversing direction for a shorter distance (0.21–0.81 Hz, Fig. 4e). As a result, hemolymph can move in two or more alternative directions at many vein junctions, and flow can appear tidal in some smaller veins (Supplementary Movie 5). In many insect wings, leading edge veins are relatively larger in diameter and tend to decrease across the span and chord (i.e., along the length and width of the wing). We found that pulsatility dominates hemolymph movement in the leading edge and trailing edges of both fore- and hindwings, where the veins are larger in diameter (170–250 μm than in the tip and lattice regions of the wing). Approximately in sync with the anterior wing heart pulse frequency of 0.64 Hz, average hemolymph flow in the forewing pulses approximately 0.56 Hz back and forth within veins (5c, Supplementary Movie 8), with net movement eventually proceeding towards the wing tip and downward through cross-veins. Pulsation at the wing hinge is irregular and subject to respiration of thoracic air sacs and is not aligned with wing pulsatility due to flow constraints in the vein network. Pulse frequency (i.e., 'pulsatility') indicates cyclic changes in hemolymph velocity, quantified by counting velocity peaks in a velocity trace (see the "Methods" section). Because we did not measure dorsal or wing heart contraction directly, we are unable to draw conclusions about correlations between the pulsatility of flow in the wing veins and the exact stroke cycles of these pumps.

Aperiodic flow occurs where particles move in one direction continuously (without stopping); velocity may increase and decrease in sync with hemolymph pulsing, but the particles never stop entirely (Fig. 3a). We observed aperiodic flow, as well as pulsatile flow, within the remaining three wing regions—the wing tip, lattice, and trailing edge regions (Supplementary Movies 4–6) (Fig. 3d, iii/viii, iv/ix, and v/x). Pulsation tends to be damped within the wing tip and lattice regions, with the flow more often moving continuously towards the trailing edge, whereas pulsatile flow is more common within the trailing edge region of both fore- and hind-wings, where hemolymph is pumped out of the wing.

Leaky flow, a flow behavior in insect wings that has been qualitatively noted before[26] but not quantified, occurs when particles move out of the wing veins and into an adjacent membranous region (a large sinus region), eventually flowing back into veins that surround the sinus (Supplementary Movies 2 and 3). In the "membrane" region of the wing (Fig. 3b, d, ii/vii), occurring at approximately two-thirds of the wing span and towards the leading edge, hemolymph flows out of the

leading edge veins (costa, subcosta, and radius) and into the pocket-like membranous sinus (Supplementary Movies 2 and 3). Hemolymph pools within this membrane-sinus of the pseudo-stigma in both wings (Supplementary Movies 2 and 3), supplied by leakiness of the veins. While exhibiting a "type" of flow, this term also reflects how regions differ structurally. Not all portions of the wing allow for leaky flow (i.e. flow in the membrane), and this depends on vein structure, and whether it is fully tubular, or shallow, u-shaped, or has pores to allow for leakage (Supplementary Movies 2 and 3). However, both pulsatile and aperiodic behaviors can occur within leaky regions (Fig. 3d, i/vi and ii/vii, Supplementary Movies 1–3). Particles moving from vein to membrane in this region exhibited velocities similar to those in the tubular veins of the leading edge.

Leakiness also occurs in the pseudo-stigmas in some other insect wings[26]. These "false sinuses" are thought to be regions of potential aerodynamic importance, where the additional mass in the leading edge may act as an "inertial regulator" of wing pitch during flapping flight[26,41]. Leakiness is not constrained to pseudo-stigmas, and may also be present in the leathery tegmen or elytra (i.e., modified forewings of beetles), where tubular veins are absent from much of the wing[7]. In contrast, dragonflies display a "true" sinus in the form of the pterostigma, a thickened, rectangular piece of cuticle near the leading edge of the wing, which forms a sinus where hemolymph pools.

**Viscous effects dominate flow regimes in the wings**. An insect wing is essentially a microfluidic device, sorting hemocytes and other hemolymph factors throughout the wing. To understand its efficiency and potential applications to bioinspired devices we calculated several key dimensionless flow parameters. We characterized flow regimes in wing regions by calculating the following dimensionless numbers per particle trajectory, and then presented averages per region: Péclet number (Pe, the ratio of advective to diffusive transport; Fig. 4d), Reynolds number (Re, ratio of inertial to viscous flows; Fig. 4f), and Womersley number (Wo, the ratio of pulsatility with respect to viscosity effects; Fig. 4g). Lastly, we measured pulse frequency (Fig. 4e) as a metric of pumping (though pumping was not measured directly) to characterize the pulsatility of flows.

Pe (Fig. 4d) is of similar magnitude between wing regions, with the exception of the trailing edge of the forewing, where it is slightly higher, and the wing tip, where it is near 1. Pulse frequency (Fig. 4e), measured as the number of velocity peaks over time, is highest in regions of the wing where flow is returning back to the body, such as the trailing edge region (0.04 and 0.03 Hz). Re (Fig. 4f) is similar between wing regions, but in the trailing edge region of the hindwing, it increases nearly an order of magnitude (lowest Re—0.01 to highest Re—0.09) where hemolymph is being pumped back into the body. Here the flow is dominated by viscous effects, and this increase compared to the rest of the wing underscores the importance of the thoracic pumping organs in driving wing circulation. Wo (Fig. 4g) is similar in all wing regions, except for a noticeable decrease in the wing tip, where flow and pulsatility tend to slow down (Fig. 5g). Compared with the human body, hemolymph flow within both wings has a similar Wo to arterioles and venules[42].

**Body flows are fastest**. Flow velocities measured within the thorax and the rest of the body were much faster than those in the wings (Fig. 5). Flows in the thorax near the wing hinge (i.e. FW and HW HGE) were irregular, with mixing of in-going hemolymph with hemolymph in the thoracic cavity. The connection between the hinge and the entry into the wing vein conduits is complex, with flow likely influenced by thoracic air sacs (hence a

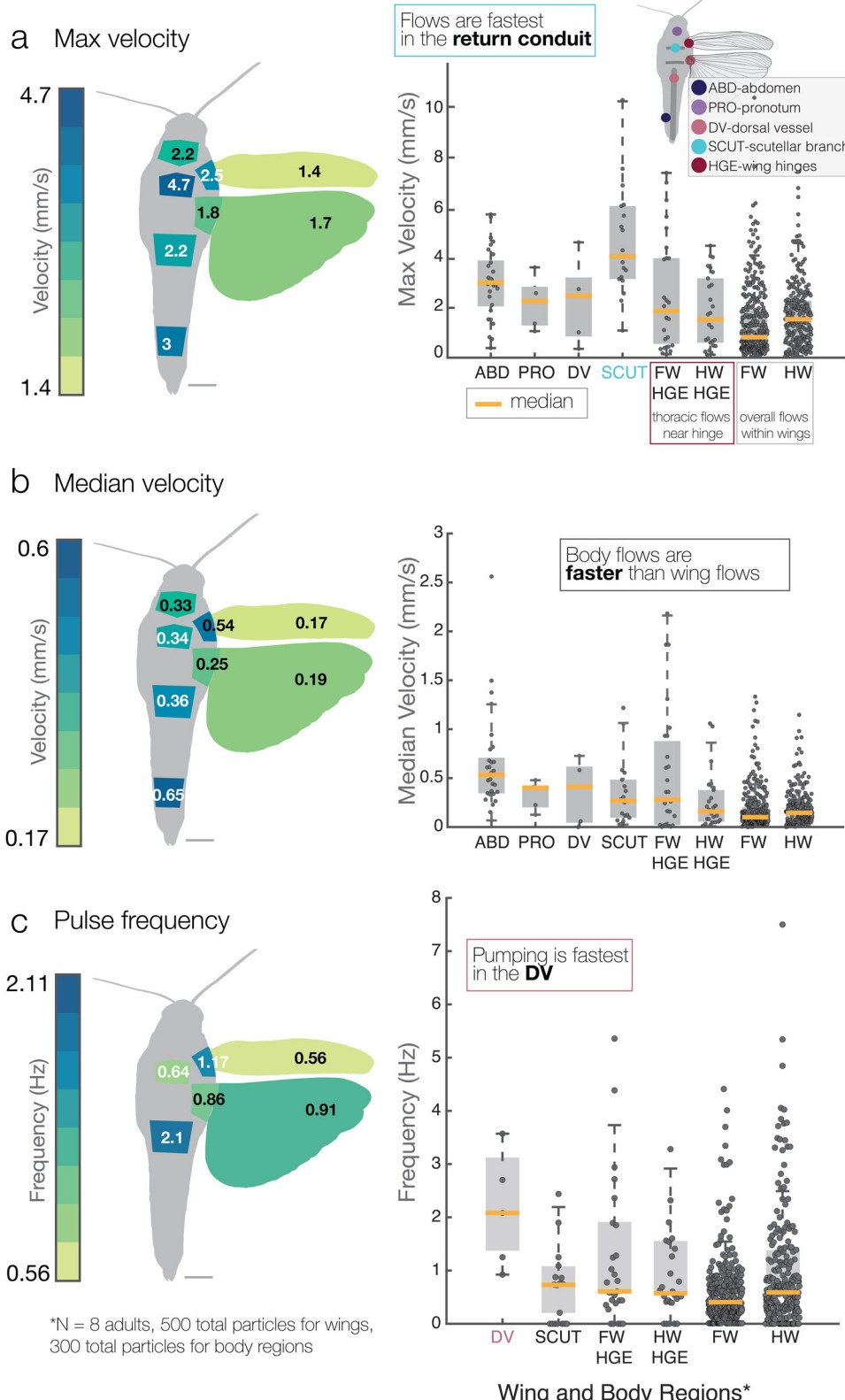

**Fig. 5 Comparison of flow in the body versus the wings. a–c** Averages of the maximum instantaneous particle velocities, instantaneous median velocities, and pulse frequencies across the body and wings, along with box and whisker plots (median in orange) of particle data (see the "Methods" section for calculation). In **a** and **b** flows are faster within the body than in wings. **c** Average pulse frequency (as calculated in Fig. 4e) shows that pumping is highest in the dorsal vessel at 2.1 Hz, versus the scutellar branch (SCUT) at 0.64 Hz (return conduit for flow) and wing areas which have similar frequencies. FW-HGE indicates thoracic flows near the hindwing hinge. Scale bar—5 mm. For each box, the central mark is the median, and the bottom/top edges indicate the 25th and 75th percentiles. Whiskers extend to extreme data points ($n = 8$ individual grasshoppers and 500 digitized particles within the wings, 300 particles within the body).

difference in pulse frequency as well). This is reflected in the high speeds of 1.8 and 2.5 mm/s seen in the wing hinge regions (Fig. 5a). The dorsal vessel and scutellar branches (Fig. 2b, Supplementary Movie 7) both displayed significantly higher flow velocities (Fig. 5a, b) than those measured in the forewing, hindwing, or hindwing hinge (paired $t$-test, $P < 0.001$). Flow velocities within the abdomen were also higher than those within the forewing, hindwing, and hindwing hinge, whereas flows in the pronotum (a protective neck collar) were not statistically different from any other sampled regions except for the scutellar branch (paired $t$-test, $P < 0.05$).

Due to the role that the pumping organs play in driving hemolymph flow, differences in pulse frequency (i.e., pulsatility) among sampled regions of the body and wings are similar to those trends seen in flow velocity (Fig. 5c). We measured a mean pulse frequency of 2.1 Hz in the dorsal vessel, which is about 3× higher than the pulse frequency of 0.64 Hz measured in the return conduits from the wing (scutellar branches). This return frequency is somewhat higher than previously measured dorsal vessel pumping frequencies in *Schistocerca* (0.92 Hz)[43], but heart rates in insects may vary with other factors such as temperature, body size, and that the insect is restrained with wings spread, so that direct comparisons across studies are not warranted. Pulse frequencies (i.e. pumping frequency) of the scutellar branch, wing hinges, and wings are not markedly different (Fig. 5c, paired $t$-test). The similarity in hemolymph pulsing among these regions is also confirmed by similar Womersley values calculated for the fore- and hindwing regions near the hinges (Fig. 4g). The averages between the wings may indicate more about structure than proximity to pumping; the hindwing has larger and longer vein conduits than the forewing's many cross veins.

**Conclusions.** Our results show that on a local level, hemolymph circulates through every vein within the wing, even the smallest cross-veins, and is identifiable as three different types of local flow behaviors: pulsatile, aperiodic, and leaky flow. With leaky flow, we discovered a surprising feature in the pseudostigma region of the wing, where hemolymph flows and moves from larger longitudinal veins, pooling into membrane sinuses (Supplementary Movies 2 and 3). This is the first work to support the qualitative evidence made by Arnold in 1963 of flow in pseudostigmas in insects[26]. Despite the straightforward pattern of circuitous flow throughout the whole wing, local flow behaviors within individual veins are complex and time-varying and are present in different combinations within each region of the wing. A complex circuitous route leads to a critical question, are these local flow patterns efficient in transporting hemolymph? The use of "efficient" requires distinction. First, these measurements were conducted in at-rest wings; flow patterns may be completely different in flight (see Wang et al., 2021[12]). Second, efficiency depends on complex physiological relationships such as the relationship between flow and the wing's nervous system, or flow and increased respiration. Exploring these relationships in further experiments could explain how wing circulatory systems differ between insects and how efficiency could vary between insects with different sensory needs (i.e., flightless but winged insects).

Future studies incorporating high-resolution x-ray tomography to visualize internal vein tissues in unprecedented detail[1] would allow for accurate physical measurements of vein structure that could be used to more precisely model the morphological network. Specifically, the tracheal network within the veins does not extend to all veins, but where present, its compression and re-inflation should influence hemolymph circulation. Some insect orders, such as lepidopterans, use tracheal expansion to promote the tidal flow of hemolymph in and out of the wings, which has

also only been measured in Lepidoptera[19]. Recently, Tsai and colleagues measured the change in channel width within the vein as tracheae expanded and contracted cyclically[8]. It is worth noting that particles and hemocytes do get stuck on tissues, and tracheae can inhibit flow (see Supplementary Movies 3 and 7 for examples), which suggests further investigation into system coupling. Future work combining fluorescent microscopy with electromyography, pressure sensors, and tracking tracheal expansion/compression would enable a more complete modeling of circulation for testing hypotheses of respiratory–circulatory coupling[44].

Additionally, little is known about how differences in body size, which extend over several orders of magnitude across insect species, and the equally wide variation in life history strategies and flight behaviors, may affect patterns of hemolymph flow within wings. For example, a migratory insect such as a monarch butterfly, which often glides along air currents and needs to maximize energetic efficiency to travel long distances without feeding, may instead benefit from slower wing hemolymph flows, which require less active pumping. In addition, flow speeds are likely to vary widely within insect orders such as Lepidoptera, which display large variations in body size and venation.

In flight, hemolymph may play another functional role; flapping induces faster hemolymph circulation within the wing[12]. Potentially, the flapping flight may influence the efficiency of the circuit, as it speeds up how quickly hemolymph reaches the wing tip (see Wang et al., 2021)[12]. As it stands, hemolymph flow in the heavily pulsatile regions observed nearest the wing hinge, may not move as quickly in grasshoppers as it may if the animal is moving. Individuals of genus *Schistocerca* employ an "umbrella-effect" during flight, in which the forewings and hindwings flap in anti-phase and the corrugated hindwings balloon out, flexibly deforming with each wing flap[45,46]. Their pseudo-stigma region also deforms in this dynamic motion and likely pressurizes fluid on either side of flexion lines. Thus this dynamic movement, similar to Wang et al.'s (2021) measurements in dragonflies[12] likely moves hemolymph throughout the circuit more quickly. So, what is the role of this fluid in flapping vs. gliding flight? Does flapping induce pulsatility everywhere across the wing? How do the local flow behaviors change as the veins deform and fold? It is likely that the mechanical behavior of insect wings is shaped not only by the wing's material properties and the pattern of supporting wing veins, but also by the presence, and perhaps movement, of hemolymph within the veins.

Essentially, an insect wing can be considered a soft-bodied microfluidic device, composed of thin membranes and tubes, which develops over time and changes shape dynamically, both during metamorphosis and in adulthood (particularly in species where the adult wings can fold). Hence the importance of characterizing this network with a set of dimensionless flow parameters. Insect wings are deployed during ecdysis[9] with the wing venation networks intact, a process that could inspire new technologies in the field of microfluidics[47]. During metamorphosis, the adult wing becomes fully formed, but it remains folded into a complex, origami-like structure that must be unfurled hydraulically during eclosion. This active process, which lasts for about 40–60 min in many insects, relies upon the network of tubular wing veins to pressurize the wing with hemolymph[10]. Relatively little is known about the diverse mechanics involved in this process (outside of *Drosophila*)[2], but the potential applications of an improved understanding of wing expansion extend from small, biomedical devices to large, autonomously unfolding satellite solar panels.

Every flight behavior performed by winged insects, from predation to pollination, relies on functioning wings. In an age of massive declines in insect populations and diversity due

to industrialization, climate change, and disease, additional investigations into the living networks within the complex, yet fragile insect wings will only benefit our understanding of the unique role that these structures play, and the external pressures that may affect their ability to function properly.

## Methods

### Animals

Schistocerca americana. Nymphs (2nd–4th instars) obtained from USDA (Sydney, Montana) were maintained at 30–35 °C (16:8 h light cycle) and reared in accordance with USDA Aphis permits (#:P526P-16-04590). Once nymphs eclosed, adults were placed in a separate enclosure. Adults were regularly fed romaine lettuce, which supplied both nutrition and water.

### Fluorescent particle injection

To prepare for fluorescent microscopy, adult *S. americana* was briefly anesthetized with carbon dioxide and placed ventral-side up. Using an insect pin, a small hole was drilled (~0.1 mm$^2$) in the second or third abdominal segment. 6–10 µl of a mixture of fluorescent green particles (Thermo Scientific; density, 1.05 g/cm$^3$; fluorescence, 589 nm) were injected using a 2.5 µm glass syringe (Hamilton Co., syringe model No. 62, Ref. 87942, Reno, NV) with a pulled borosilicate capillary tube as a needle (Fig. 2a). This mixture contained neutrally buoyant polystyrene particles of sizes 3 and 6 µm.

The mixture allowed the flow to be observed at both large and small focal distances (Fig. 2). After injection, *S. americana* was quickly restrained with modeling clay, allowing the live insect to remain at rest without moving or causing self-harm. Fore- and hindwings were spread and sandwiched between two glass slides (7.5 × 5 cm) in a planar position, which simulated a wing-extended flight posture and allowed for visualization of hemolymph flow (Fig. 2). Due to the insect's open circulatory system (Fig. 1b, c), injected particles flowed readily with hemolymph and were observed moving in and out of pumping organs, the body, and appendages[21]. Because some particles became stuck on tissues within the body and within the wings, we only measured particles that were clearly freely moving with the flow, which also can be seen moving with hemocytes (Supplementary Movie 1). We did measure slowing particles and those that reversed direction, reflecting pulsatility in the flow. To avoid unduly stressing a grasshopper, experiments were performed 5–10 min after injection of particles, for 3–4 h.

### Flow visualization

Particle movement was captured in eight adult *S. americana* (~3–5 months old) on a fluorescent microscope (Zeiss AxioZoom V16 Zoom, using Zeiss software) at the Harvard Center for Biological Imaging (Cambridge, MA). Due to focal constraints, particle sizes, and clarity of the venation, no more than a third of the wing could be viewed at a time (~300 mm$^2$ for the hindwing and 10–50 mm$^2$ for the forewing). Thus, movies were captured in a tiled fashion across the span and chord of the wing (Fig. 2c), imaged sequentially from the wing base to the wing tip. In theory, a particle could be followed from the wing hinge to the wing tip and back again, but in practice, visibility, frame rate, and file-saving time limited the tracking distance of individual particles to smaller sections within the wing. Frame rates ranged from 10 to 100 frames per second, where higher frame rates were necessary to resolve rapid flow at the wing hearts and leading edge veins.

### Particle tracking

The instantaneous position of ~800 particles from 228 recordings of 8 individual adult grasshoppers was identified and quantified for calculations of velocity. The wing was divided into sections: leading edge, membrane, wing tip, lattice, and trailing edge. These were based on vein arrangement and allowed for averaging data for specific wing regions. The membrane region exists within the leading edge but is structurally remarkable due to the pooling of hemolymph in that membrane sinus. We used two methods to track particles. First, manual and semi-automated tracking was conducted using a MATLAB-based point-tracking program (DLTdv5)[36]. Second, we employed custom multi-parametric particle tracking algorithms adapted from previous work[48–51]. Background subtraction was first applied to each frame in the time series to address low contrast ratios and compensate for uneven spatial illumination levels. A frame-wise linear intensity adjustment was applied, such that 1% of the total pixels were saturated, accounting for temporal fluorescence decay due to photobleaching. A local Hessian matrix of the intensity was calculated for each pixel, and the particles were marked by negative λ2 values in the Hessian eigenmaps. A dynamic erosion procedure with an adaptive threshold was used to identify each intensity peak of all particles that were analyzed. Subsequently, a dilation procedure was used to expand the boundaries from the identified peaks until it captured the course boundary of each particle. Finally, the coarse segmentation was mapped back to the original resolution and refined. The refining expansion stopped either when the pixel intensity fell below 25% of the peak intensity within the particle, or when it met the edges detected by a Canny filter[52]. This algorithm identifies the most probable correspondence between particles by taking into consideration the characteristics of each particle (brightness, area, diameter, and orientation) in addition to the classic nearest-neighbor criterion as tracking parameters. Flow data and all corresponding data movies are available upon request.

### Flow calculations

Particle trajectories were identified, placed on a normalized wing coordinate system, and categorized into five wing regions (Fig. 3b, c) and major body regions (Fig. 2c). Trajectories with less than 25 data points were removed from the main dataset. Velocity data were smoothed in MATLAB using a moving-mean function (Matlab movmean) with a window length of 5. Across the five wing regions (Fig. 4), instantaneous velocity (max and median), average vein radii, pulse frequency, Péclet number, Reynolds number, Pulse frequency, and Womersley number were calculated. Instantaneous velocity ($V_{instant}$, mm/s) quantifies how fast particles moved through a region; maximum and median velocities were also calculated to show a range of particle movement (over instantaneous time). Vein radius was determined by taking an average of 25 vein diameters within a wing region. Pulse frequency ($f$, Hz) measures flow pulsatility, where the number of peaks in a velocity trace (over time) was used as an indication of periodicity (see example plot in Fig. 4e). To identify peaks, velocity traces were normalized by max velocity, and then peaks were detected using the MATLAB function *findpeaks* and a threshold value of 0.3, which captured most apparent pulsatility. Péclet number reflects the ratio of viscous flows to diffusive transport. Reynolds number indicates the ratio of inertial to viscous fluid forces. Womersley number detects the relevance of pulsatility to the viscous effects in a flow. The equations used are as follows:

$$V_{instant} = \frac{\sqrt{(x_i - x_{i+1})^2 + (y_i - y_{i+1})^2}}{time_i} \tag{1}$$

$$Pe = \frac{rV_{max}}{D_{O2}} \tag{2}$$

$$Re = \frac{\rho V_{max} r}{\mu} \tag{3}$$

$$Wo = r \left(\frac{f\rho}{\mu}\right)^{0.5} \tag{4}$$

where points $(x_i, y_i)$ and $(x_{i+1}, y_{i+1})$ are the instantaneous trajectory points through which a given particle travels, $time_i$ is the instantaneous time interval in seconds, $\rho$ is the density of water, $\mu$ is the dynamic viscosity of water, $r$ is the average radii per wing region (values found in code online), $f$ is the average pulse frequency per wing region, and $D_{O2}$ is the diffusion coefficient of oxygen in water at $0.000018*(1*10^{-4})$. For the dynamic viscosity of hemolymph, we used 0.0010518 Pa s at 18 °C.

### Reporting summary

Further information on research design is available in the Nature Portfolio Reporting Summary linked to this article.

## Data availability

Data movies are available upon request. Tracking data are available at https://doi.org/10.5281/zenodo.7637483.

## Code availability

Code used in data analysis was previously written[48–51]. Matlab analysis code, tracking data, and parameter files can be found at https://github.com/maryksalcedo/wingflow_grasshoppers.git.

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

## Acknowledgements

We thank Dr. Stefan Jaronski (USDA) for his continuous grasshopper supply, thoughtful discussions, and support, and Dr. Missy Holbrook for helpful advice in finding a model system. We thank the Holbrook lab for giving space to the grasshopper colony for 2 years and allowing insect experiments in her plant lab. We thank L. Mahadevan for his advice, comments, and edits throughout the experiment and process. We thank the Harvard Center for Biological Imaging for infrastructure and support, specifically Dr. Doug Richardson for his advice and time training. We thank Dr. Siddarth Srinivasan for his expertise in measuring biological flows and his time spent in helping train on the microscope. We thank the Socha Lab members for their thoughtful feedback and support in analysis and writing. Lastly, special thanks to Dr. Jacob Peters for his advice and support throughout the project. This research was funded through two US National Science Foundation (NSF) fellowships to M.K.S. (NSF GRFP and an NSF PRFB 1812215) and partially supported by NSF 1558052 to J.J.S. M.K.S. was also partially supported by the United States of Agriculture NIFA Fellowship (Award: 2022-67012-37679).

## Author contributions

M.K.S. conceived of the research project, collected and analyzed data, and wrote the manuscript. B.H.J. analyzed data, added methods, and helped to edit the manuscript. P.P.V. contributed methods and manuscript edits. S.A.C., J.J.S., and N.E.P. contributed significant edits and advice throughout the project.

## Competing interests

The authors declare no competing interests.
