## [Peer Review File · Communications Biology]

Reviewers' comments:

Reviewer #1 (Remarks to the Author):

This is certainly a very interesting study, which deserves publication with only minor changes. The paper is very well written. The presentation is very nice. And the paper certainly contributes to the field and to our understanding of the wing biomechanics. The study is novel and can be potentially of interest of anyone interested in wing biomechanics and insect flight.

- Introduction: I appreciate the detailed, interesting introduction. I like the introduction very much and, hence, what I mention here is a suggestion that can be taken or not. My point is whether introduction adequately reflects the importance of hemolymph. I am uncertain whether, after reading the introduction, a non-specialist would find out why we need to know more about the hemolymph flow in the wing. To achieve this, the authors would need to add a few sentences (perhaps to the first paragraph and after the one last sentence) indicating what would happen to the wing (particularly the wing properties) if the wing dries out.

- Line 80: "hemolymph flows through all veins in each wing"
Shouldn't this be "hemolymph flows through all longitudinal veins in each wing"?

- Line 84: "u-shape" veins
This is difficult to imagine.

- Fig. 2C
Is there any reason that the mapped particles across the body are clustered in some regions?

- Line 91: "where veins are most likely to be damaged.[38]"
Although [38] is certainly an amazing study, there are a few other more specialized studies in this area including a review article by my colleagues and me (Rajabi et al., 2020, Journal of Experimental Biology, jeb215194). If you would like you could cite that here.

- Line 101
"with" should be removed.

- I wonder whether particle velocity can accurately represent the hemolymph velocity. I ask particularly because the inner surface of veins are not smooth and particles can stuck in such roughnesses, while this wouldn't be the case for hemolymph. I appreciate that this is one of the few feasible ways to measure hemolymph flow, but wonder whether this needs an explanation.

- The whole paragraph starting from L113
This seems a very inefficient way of transportation, isn't it?

- Line 124 "Within the wing, hemolymph flows faster in regions near the body and slower in regions toward the tip of the wing."
Could this imply that some wing parts might be more hydrated than others?

- Since I started reading the paper, there were two questions that wanted to know their answers. I think that I got the answer to the questions to some extent. However, I hope these could be a bit more highlighted in the revision. I believe that everyone, at least those within the community, very well know that there's hemolymph flowing in the wings. However, I don't think that everyone knows the answer to these questions: (1) whether hemolymph flows within the membrane and (2) whether hemolymph flows within the cross veins. After reading the paper, I can say that hemolymph flows within some cross veins and membranes, but not all. Am I right?

- Conclusions
OK! Now I got the answer to one of my questions in the previous comment: All cross veins! I believe it is worth to mention something about hemolymph in membranes here too.
Very beautiful conclusion!

Reviewer #2 (Remarks to the Author):

This manuscript describes hemolymph circulation in an insect of great agricultural importance: the locust, *Schistocerca americana*. The authors use fluorescence microscopy to visualize flow in the wings, describe the path and forms of flow, quantify the velocity of flow, and compare flow in the wings to flow in the dorsal vessel and hemocoel. This study builds on prior work conducted in the mosquito (a smaller insect with a much simpler wing) but this study goes beyond that prior work by studying a much more complex wing and computing additional parameters related to fluid dynamics. Overall, I am very supportive of this manuscript, as it is a significant advancement in an area of insect biology that, although essential for many physiological processes, has received little attention. The study is well designed, and I do not have any specific criticism. I have several questions and points for clarification, all of which can be addressed without additional experimentation.

1. Consider a different heading to the first section of the results. The finding that flow occurs in all wing veins cannot be surprising. This section can be used to better describe the wing and introduce some of the nomenclature of the veins that are later invoked in the quantitative sections.
2. In several places, the diameter of a vein is detailed, and considered in terms of how it impacts hemolymph flow. However, one aspect that does not seem to be considered is the area within the vein that is available for flow. Inside these veins are trachea, and according to figure 1, the trachea occupy the majority of the veinal space. Is the diagram in figure 1 proportionally accurate? Is it possible then, that the diameter of a fluorescent particle impacts its velocity because of how it squeezes between the vein wall and the trachea? From looking at the first movie, it appears that the smaller particles are moving much faster than the larger particles.
3. The velocities being presented are maximum and median. I have a couple of questions. First, what is the distance (total and effective distance) and time that these particles are being tracked. I ask because of the pulsatile flow observed. Oscillation can have particles moving very fast but not moving very far. See the first video, where the particles are moving quite fast, but not accomplishing much in traversing the circuit.
4. It is interesting that pulsatile flow is more pronounced in the leading edge. Not having seen the videos, I would have predicted the opposite: more pronounced pulsatile flow in the trailing edge because it is closer to the wing heart. Can the authors speculate on why this happens? If it is because of the radius of the vein, does this consider the diameter of the trachea that lies within?
5. In figure 3A, more clarity on the plot is necessary. Is this one particle? Surely not because there is a standard deviation. If not, then how is it that particles are so coordinated between individuals? For example, see the aperiodic jump at 30 sec. Could this be all the particles in one insect? How does it vary between insects?
6. Supplemental graphs that show the mean velocity and standard deviation (or SEM) of all particles tracked would improve the manuscript. As is, the data on the different parts of the wing are highly summarized. See figure 5, where data presentation is excellent.
7. The section "viscous effects..." lacks context. The parameters calculated need to be better defined and explained. More value is needed in the "why should the reader care?" category. I am not questioning the importance. I am suggesting that it needs to be better highlighted.
8. I am a little confused as to "pulse frequency". Is this the same as dorsal vessel or wing heart contraction? Or is it the pulsation of flow? Could contractions be observed (which would be more informative)? What accounts for the difference between FW HGE and FW? Shouldn't they be the same?
9. In this study, data on all particles tracked within an individual were pooled. Were there differences between the 8 insects assayed?

10. The statement in line 190 "measurements of pumping frequency may vary depending on the specific location sampled along the dorsal vessel" needs a reference. As far as I know, a dorsal vessel contraction in the abdomen is a wave-like contraction that travels across the entire length of the heart (abdominal portion of the dorsal vessel). Therefore, the pulsation frequency should be the same regardless of location in the abdomen.

Reviewer 1:

This is certainly a very interesting study, which deserves publication with only minor changes. The paper is very well written. The presentation is very nice. And the paper certainly contributes to the field and to our understanding of the wing biomechanics. The study is novel and can be potentially of interest of anyone interested in wing biomechanics and insect flight.

Author response: Thank you for the thoughtful comments!

1. Introduction: I appreciate the detailed, interesting introduction. I like the introduction very much and, hence, what I mention here is a suggestion that can be taken or not. My point is whether introduction adequately reflects the importance of hemolymph. I am uncertain whether, after reading the introduction, a non-specialist would find out why we need to know more about the hemolymph flow in the wing. To achieve this, the authors would need to add a few sentences (perhaps to the first paragraph and after the one last sentence) indicating what would happen to the wing (particularly the wing properties) if the wing dries out.

Author response: We have added emphasizing sentences in the first paragraph, “These systems extend and branch into the wing as well, necessitating nourishment by hemolymph as in the body,” and “In fact, under desiccation, insect cuticle dramatically decreases in toughness.”

2. Line 80: “hemolymph flows through all veins in each wing” - Shouldn't this be “hemolymph flows through all longitudinal veins in each wing”?

Author response: We did mean that hemolymph flows through “all” veins, but to clarify we adjusted the sentence to say, “Results from these techniques indicate that hemolymph flows through all longitudinal and cross veins, in each wing.”

3. Line 84: “u-shape” veins: This is difficult to imagine.

Author response: Here we were trying to identify a region of the forewing near the base of the wing in which veins are dense and interconnected, but not truly circular. We adjusted the sentence as follows: “Particularly near the wing base in the leading edge region of the forewing, the wing veins are not uniformly circular in cross-section, but appear flattened and shallow, interconnected by many S-shaped cross-veins. Particles in this region are seen flowing alongside hemocytes.”

4. Fig. 2C: Is there any reason that the mapped particles across the body are clustered in some regions?

Author response: Clustering of the particles indicates that multiple particles were tracked within that region and jittered to show number/density. We added these sentences in the first paragraph of the Results to clarify those features in the figure: “In the wing

schematics of Fig. 2C, particle location indicates the normalized starting position of tracking, and clustering indicates that multiple particles were tracked within a region. In Fig. 2C (bottom), the particles within the body are jittered and represent a general tracking location.”

5. Line 91: “where veins are most likely to be damaged.[38]”

Although [38] is certainly an amazing study, there are a few other more specialized studies in this area including a review article by my colleagues and me (Rajabi et al., 2020, Journal of Experimental Biology, jeb215194). If you would like you could cite that here.

Author response: We have added the suggested citation.

6. Line 10: “with” should be removed.

Author response: “With” has been deleted.

7. *I wonder whether particle velocity can accurately represent the hemolymph velocity. I ask particularly because the inner surface of veins are not smooth and particles can stuck in such roughnesses, while this wouldn't be the case for hemolymph. I appreciate that this is one of the few feasible ways to measure hemolymph flow, but wonder whether this needs an explanation.*

Author response: We added a sentence to the methods, “Because some particles became stuck on tissues within the body and within the wings, we only measured particles that were clearly freely moving with the flow, which also can be seen moving with hemocytes (Supp. Movie 1). We did measure slowing particles, and those that reversed direction, reflecting pulsatility in the flow. To avoid unduly stressing a locust, experiments were performed 5-10 minutes after injection of particles, for 3-4 hours.”

This point is reiterated at the beginning of our Results section too, “Using neutrally buoyant fluorescent particles and high-speed video, we recorded particles flowing in sync with hemolymph and hemocytes that are advected into the wing at the wing base (Supp. Movie 1). “

8. *The whole paragraph starting from L113: This seems a very inefficient way of transportation, isn't it?*

Author response: We added sentences in the Conclusion section to get at this point, “A complex circuitous route leads to a critical question: Are these local flow patterns efficient in transporting hemolymph? The use of “efficient” requires distinction. First, these measurements take place in at-rest wings; flow patterns may be completely different in flight. Second, efficiency depends on complex physiological relationships such as the relationship between flow and the wing's nervous system, or flow and

increased respiration. Exploring these relationships in further experiments could explain how wing circulatory systems differ between insects and how efficiency could vary between insects with different sensory needs (i.e., flightless but winged insects).”

9. Line 124 “*Within the wing, hemolymph flows faster in regions near the body and slower in regions toward the tip of the wing.*” *Could this imply that some wing parts might be more hydrated than others?*

Author response: We added a sentence, “These higher speeds may reflect proximity to the main pumping organs, or perhaps reflect functional requirements for greater hydration at the wing hinge, correlating with the thick layers of resilin (Kovalev et al 2018).”

10. *Since I started reading the paper, there were two questions that wanted to know their answers. I think that I got the answer to the questions to some extent. However, I hope these could be a bit more highlighted in the revision. I believe that everyone, at least those within the community, very well know that there’s hemolymph flowing in the wings. However, I don’t think that everyone knows the answer to these questions: (1) whether hemolymph flows within the membrane and (2) whether hemolymph flows within the cross veins. After reading the paper, I can say that hemolymph flows within some cross veins and membranes, but not all. Am I right?*

Author response: Yes, we agree. To address your points, we added notes throughout the manuscript to emphasize the hemolymph flow in the membrane, especially in the section, “Specific flow behaviors: pulsatile, aperiodic, and leaky.”

11. *Conclusions: OK! Now I got the answer to one of my questions in the previous comment: All cross veins! I believe it is worth to mention something about hemolymph in membranes here too. Very beautiful conclusion!*

Author response: We appreciated this feedback and added earlier comments that hemolymph is found in the cross veins too. New sentences in conclusion: “With leaky flow, we discovered a surprising feature in the pseudostigma region of the wing, where hemolymph flows and moves from larger longitudinal veins, pooling into membrane sinuses (Supp. Movie 2 and 3).”

Reviewer 2:

*This manuscript describes hemolymph circulation in an insect of great agricultural importance: the locust, *Schistocerca americana*. The authors use fluorescence microscopy to visualize flow in the wings, describe the path and forms of flow, quantify the velocity of flow, and compare flow in the wings to flow in the dorsal vessel and hemocoel. This study builds on prior work conducted in the mosquito (a smaller insect with a much simpler wing) but this study goes beyond that prior work by studying a much more complex wing and computing additional parameters related to fluid dynamics. Overall, I am very supportive of this manuscript, as it is a significant advancement in an area of insect biology that, although essential for many physiological processes, has received little attention. The study is well*

designed, and I do not have any specific criticism. I have several questions and points for clarification, all of which can be addressed without additional experimentation.

Author response: Thank you as well for the thoughtful comments!

1. Consider a different heading to the first section of the results. The finding that flow occurs in all wing veins cannot be surprising. This section can be used to better describe the wing and introduce some of the nomenclature of the veins that are later invoked in the quantitative sections.

Author Response: We appreciate this sentiment that perhaps our result is not surprising, and wish it were so! However, after extensive work in this area and talking to many biologists—including entomologists—we conclude that many people are not aware that there is hemolymph flow in all of the wing veins. Reviewer 1’s comments reflect our view: see their comments in #10 and #11.

We explain much of our nomenclature in the second section, “Local flow behaviors depend on wing location.” However, toward addressing this comment, we added these sentences in the “Particle Tracking” of the methods to describe the regions: “The wing was divided into sections: leading edge, membrane, wing tip, lattice, and trailing edge. These were based on vein arrangement and allowed for averaging data for specific wing regions. The membrane region exists within the leading edge, but is structurally significant due to pooling of hemolymph in that membrane sinus.”

2. In several places, the diameter of a vein is detailed, and considered in terms of how it impacts hemolymph flow. However, one aspect that does not seem to be considered is the area within the vein that is available for flow. Inside these veins are trachea, and according to figure 1, the trachea occupy the majority of the veinal space. Is the diagram in figure 1 proportionally accurate? Is it possible then, that the diameter of a fluorescent particle impacts its velocity because of how it squeezes between the vein wall and the trachea? From looking at the first movie, it appears that the smaller particles are moving much faster than the larger particles.

Author Response: We added a correction to the Figure 1, Part C figure, “An example cross-section (not proportional)” to indicate it is more of a schematic. However, our work with fresh wing tissue at the synchrotron light source at Argonne National Laboratory, has shown similar proportions of hemolymph to tracheae. Tracheae can occupy much of the space, and hemolymph can move around it, especially as tracheae inflate/deflate. Movie 1, particle size does matter, as large particles can get stuck on tissues. We added additional context about tracheae to the 4th introductory paragraph, indicating that it can be found within veins but not all veins.

In the first paragraph of the Results section: “Particles that became stuck on tissues were not tracked, but rather those moving freely with flow.”

Lastly, in the conclusions section, we added additional comments on the potential tracheal-hemolymph system: “Future studies incorporating high-resolution x-ray tomography to visualize internal vein tissues in unprecedented detail would allow for accurate morphological measurements of vein structure that could be used to more precisely model the network. Specifically, the tracheal network within the veins does not extend to all veins, but where present, its compression and re-inflation should influence hemolymph circulation. Some insect orders, such as lepidopterans, use tracheal expansion to promote tidal flow of hemolymph in and out of the wings, which has also only been measured in Lepidoptera. Recently, Tsai and colleagues measured the change in channel width within the vein as tracheae expanded and contracted cyclically. It is worth noting that particles and hemocytes do get stuck on tissues, and tracheae can inhibit flow (see Supp. videos 3 and 7 for examples), which suggests further investigation into system-coupling. Future work combining fluorescence microscopy with electromyography, pressure sensors, and tracking of tracheal expansion/compression would enable more complete modeling of circulation for testing hypotheses of respiratory-circulatory coupling.” (citations are in the main text)

3. The velocities being presented are maximum and median. I have a couple of questions. First, what is the distance (total and effective distance) and time that these particles are being tracked. I ask because of the pulsatile flow observed. Oscillation can have particles moving very fast but not moving very far. See the first video, where the particles are moving quite fast, but not accomplishing much in traversing the circuit.

Author Response: We appreciate this comment since we neglected to indicate that we were measuring mean instantaneous velocity, even though it was labeled in the methods at the back of the document. We measured mean instantaneous velocities for the entire particle trajectory, and reported maximum and median velocities for a given particle path. Tracked particles with less than 25 data points were removed from the data set. We have added wording in the main text and in the methods to clarify. We have also added a figure to the supplement showing all max, mean, median particle velocities within the wings.

We also added into the Results section, “To analyze hemolymph velocity at different locations, we calculated instantaneous maximum and median particle velocities across the five wing regions (Fig.4 A,B, see methods for calculations).”

At one iteration of this manuscript, there was an additional figure that had the ratio of the total distance traveled, over the effective distance, normalized by number of points

tracked, and we termed it “Path Sinuosity.” In essence it showed a similar story to the “Pulse frequency” where high path sinuosity was reflective of high pulsatility, so we removed the plot.

This comment is also interesting, as it goes back to “efficiency” of the circuit. Thus we expanded the conclusions section to discuss the pulsatile region as well.

“In flight, hemolymph may play another functional role; flapping induces faster hemolymph circulation within the wing (Yang et al. 2021). Potentially, flapping flight may influence the efficiency of the circuit, as it speeds up how quickly hemolymph reaches the wing tip (see Yang et al. 2021). As it stands, hemolymph flow in the heavily pulsatile regions observed nearest the wing hinge, may not move as quickly in locusts as it may if the animal is moving. Individuals of genus *Schistocerca* employ an "umbrella-effect" during flight, in which the forewings and hindwings flap in anti-phase and the corrugated hindwings balloon out, flexibly deforming with each wing flap. Their pseudo-stigma region also deforms in this dynamic motion, and likely pressurizes fluid on either side of flexion lines. Thus this dynamic movement, similar to what Yang et al. (2021) measured in dragonflies, likely moves hemolymph throughout the circuit more quickly. So, what is the role of this fluid in flapping vs. gliding flight? It is likely that the mechanical behavior of insect wings is shaped not only by the wing's material properties and the pattern of supporting wing veins, but also by the presence, and perhaps movement, of hemolymph within the veins.”

4. It is interesting that pulsatile flow is more pronounced in the leading edge. Not having seen the videos, I would have predicted the opposite: more pronounced pulsatile flow in the trailing edge because it is closer to the wing heart. Can the authors speculate on why this happens? If it is because of the radius of the vein, does this consider the diameter of the trachea that lies within?

Author Response: You are correct in your assumption, flow is definitely more pronounced in the trailing edge, and has higher pulse frequency and higher max and median velocities (Figure 4, A, B, E). We note in the third paragraph of the “Local flow behavior” section that both the leading and trailing edges of the forewings have wide diameter veins; however, this fact is also true of the hindwing (we added a note of this).

To emphasize this point more clearly, we also added it to the “Specific flow” section where we discuss pulsatility: “We found that pulsatility dominates hemolymph movement in the leading and trailing edges of both fore- and hindwings, where the veins are larger in diameter (170-250 microns).”

However, flow in the leading edge is quite pulsatile because hemolymph is being influenced by tracheal air sacs and hemolymph from the thoracic cavity. We added

further clarification in the Discussion to discuss studies of tracheal expansion in butterfly wing veins. Pulsatility does not depend on the radius of the vein per se, or the tracheae within it, but rather proximity to the body.

5. In figure 3A, more clarity on the plot is necessary. Is this one particle? Surely not because there is a standard deviation. If not, then how is it that particles are so coordinated between individuals? For example, see the aperiodic jump at 30 sec. Could this be all the particles in one insect? How does it vary between insects?

Author Response: We adjusted our language to indicate that Figure 3A is an example plot of particles within one region of one insect. Since no particle takes the exact same path within a vein, a generalized analysis between the locusts was necessary. It showed not only that particles fill all veins, demonstrating that the flow network is fully connected, but that local flow behaviors were similar too. Also, to avoid unduly stressing a locust, experiments were performed 5-10 minutes after injection of particles, for 3-4 hours (this info has been added to the methods).

We added sentences to Figure 3A: “We use multi-parametric particle tracking to detect bulk movement of particles which allows for tracking of large numbers of particles. Flow near the wing base (top) shows distinct pulsatility (across individuals), whereas in regions such as the wing tip (bottom), flow traverses more junctions, and patterns are less periodic (aperiodic jumps indicate travel between cross veins to long veins). These example plots show hundreds of particles within a region for one insect (both top and bottom). However, the local flow behavior is generalizable across individuals.”

We added additional clarification in the first paragraph of the Results: “Shown in Figure 3A is the algorithm tracked and calculated mean instantaneous velocity of all of the visible particles on a series of images measured over time (for one example individual). Particle movements generally appear more coordinated in regions where pulsatility is more dominant (i.e. wing base, Fig. 3A, top), compared to where it is not (i.e. wing tip, Fig. 3A, bottom).”

6. Supplemental graphs that show the mean velocity and standard deviation (or SEM) of all particles tracked would improve the manuscript. As is, the data on the different parts of the wing are highly summarized. See figure 5, where data presentation is excellent.

Author Response: We appreciate that feedback. The plots in Figure 5 do represent all tracked particles, for both the body and the wings, and we have added a note to indicate this. However, we have included plots in the back of the supplement indicating the max, mean, and median particle velocities across the wings.

[Caption] **Supplementary Figure 1.** Mean, median, and maximum instantaneous velocities across wing regions. The three above plots describe velocities of approximately 500 particles tracked across 8 insects. Calculation and tracking methods are described in the main paper.

7. The section “viscous effects...” lacks context. The parameters calculated need to be better defined and explained. More value is needed in the “why should the reader care?” category. I am not questioning the importance. I am suggesting that it needs to be better highlighted.

Author Response: We appreciate that necessary context and added sentences “An insect wing is essentially a microfluidic device, sorting hemocytes and other hemolymph factors throughout the wing. To understand its efficiency and potential applications to bioinspired devices, we calculated several key dimensionless flow parameters.” We also further reflected that in the Conclusion.

8. I am a little confused as to “pulse frequency”. Is this the same as dorsal vessel or wing heart contraction? Or is it the pulsation of flow? Could contractions be observed (which would be more informative)? What accounts for the difference between FW HGE and FW? Shouldn't they be the same?

Author Response: We clarified this in the “Specific Flows” section with, “Pulse frequency (i.e., ‘pulsatility’) indicates cyclic changes in hemolymph velocity, quantified by counting velocity peaks in a velocity trace (see Methods). Because we did not measure dorsal or wing heart contraction directly, we are unable to draw conclusions about

correlations between pulsatility of flow in the wing veins and stroke cycles of these pumps.”

Forewing Hinge Flow (FW HGE) occurs at the wing hinge, in the opening between the body and the wings. There’s lots of mixing that occurs there, hence it looks different than when it enters the conduits of wing veins. This is an important point, so we further addressed it with this sentence, “Flows in the thorax near the wing hinge (i.e., FW and HW HGE) were irregular, with mixing of in-going hemolymph with hemolymph in the thoracic cavity. The connection between the hinge and the entry into the wing vein conduits is complex, with flow likely also influenced by thoracic air sacs (hence a difference in pulse frequency as well).”

9. *In this study, data on all particles tracked within an individual were pooled. Were there differences between the 8 insects assayed?*

Author Response: In order to gather enough data, individual differences were not examined, with our main objective being general trends. Particles can take many different paths per wing, hence we pooled data. Experiments were also limited to 3-4 hours to prevent over-exhausting the insect (added to methods section).

10. *The statement in line 190 “measurements of pumping frequency may vary depending on the specific location sampled along the dorsal vessel” needs a reference. As far as I know, a dorsal vessel contraction in the abdomen is a wave-like contraction that travels across the entire length of the heart (abdominal portion of the dorsal vessel). Therefore, the pulsation frequency should be the same regardless of location in the abdomen.*

Author Response: We deleted this confusing statement and corrected it to indicate our measurement of “pulse frequency” is different from pumping frequency (as discussed in a previous response). We wrote, “We measured a mean pulse frequency of 2.1 Hz in the dorsal vessel, which is about 3x higher than the pulse frequency of 0.64 Hz measured in the return conduits from the wing (scutellar branches). This return frequency is somewhat higher than previously measured dorsal vessel pumping frequencies in *Schistocerca* (0.92 Hz), but heart rates in insects vary with factors such as temperature and body size, so direct comparisons across studies are not warranted.” Further, “pulse frequency” has been clarified earlier in the text, hopefully making this section understandable. (citations are in main text)

REVIEWERS' COMMENTS:

Reviewer #2 (Remarks to the Author):

This is an excellent manuscript. My comments have been addressed.